# Healthy homes: Stakeholder perspectives on housing interventions to reduce environmentally mediated infections

**Matthias Acklin**[1], **Jay P. Graham**[2], **Jade Benjamin-Chung**[3,4*]

**1** Department of Health Sciences and Technology, Swiss Federal Institute of Technology Zurich (ETH Zurich), **2** Division of Environmental Health Sciences, University of California, Berkeley School of Public Health, **3** Department of Epidemiology and Population Health, Stanford University, **4** Chan Zuckerberg Biohub, San Francisco

* jadebc@stanford.edu

## Abstract

Housing conditions are intrinsically linked to human health, with inadequate housing potentially increasing exposure to environmentally mediated pathogens. Beyond efforts to improve water and sanitation and reduce household air pollution, housing improvements remain relatively under-explored as health interventions. This study explored facilitators of and barriers to funding, implementing, and scaling up of housing improvements as health interventions to reduce environmentally mediated infectious diseases. Sixteen key informants (KIs) with direct experience conducting housing interventions with a goal to reduce environmentally mediated infectious diseases in low- and middle-income countries were interviewed using a semi-structured interview format. KIs had diverse backgrounds, including academics researching housing interventions, housing policy advisors, and practitioners imple-menting housing interventions. A thematic analysis approach was used to identify key themes in interview transcripts, highlighting patterns, commonalities, and variations in participants' responses. KIs emphasized that housing interventions can deliver across a broad set of health outcomes, including physical and mental health, as well as environmental, social, and economic dimensions. Funding and financial mecha-nisms to address housing interventions were highlighted as key barriers, alongside the need to provide more rigorous evidence and cost-benefit analyses for housing interventions. KIs indicated that funding limitations were likely driven by a deficiency in awareness regarding the significance of housing among decision-makers, and sug-gested that efforts are needed to foster more intersectoral collaboration. The inter-views also revealed a need for more context-specific housing policies and a need to contextualize interventions to their specific setting in order to foster community involvement and successful implementation and scale-up. Housing interventions play a pivotal role in mitigating many environmentally mediated diseases. By integrating these interventions with existing programs, such as water and sanitation or efforts to

**Data availability statement:** Excerpts from interviews will be made available upon reason-able request (matthias.acklin@hispeed.ch).

**Funding:** The author(s) received no specific funding for this work.

**Competing interests:** The authors have declared that no competing interests exist.

reduce household air pollution, there is the potential to create a more comprehensive approach to healthy housing in the face of climate change.

## Background

Housing is a fundamental component of our living environment, providing shelter and space for essential services [1]. Improvements in housing can exert an influence on public health, with the potential to decrease morbidity and mortality, improve quality of life and child well-being, alleviate socioeconomic disparities, and contribute to climate change resilience [2,3]. Housing is linked to several of the Sustainable Development Goals (SDGs), such as those addressing safe and affordable housing, infectious diseases, as well as non-communicable diseases [4].

Globally, the United Nations estimates nearly one billion people live in makeshift structures considered inadequate; that number is estimated rise to 3 billion people by 2030 [5]. Across 64 emerging economies, researchers estimated that 48% of households lacked construction with durable materials (defined as burnt bricks, tiles, shingles, iron sheets, or polished wood) [5]. Nearly half of sub-Saharan Africa's urban population resided in substandard housing in 2015 [6], and in many countries where national survey data are available, a high prevalence have dirt or sand floors: Guatemala (29.5% in DHS 2014–15); Haiti (33.8% in DHS 2016–17); Timor-Leste (51.2% in DHS 2016) [7]. Housing-related factors that can affect disease risk are 1) building structure, 2) water supply, 3) sanitation, 4) refuse storage and collection, 5) food storage and preparation, 6) cooking and heating fuels and infrastructure, 7) crowding, and 8) diseases associated with the geographical location of the housing [1,8,9]. Substandard building structures where floors, walls, and roofs are primarily constructed from unfinished materials such as earth, sand, dung, mud, and thatch can facilitate the survival of pathogens in the home and make it more difficult to remove pathogens. Additionally, an unfinished roof or walls allow vectors to enter the home [6]. Thus, housing has a link to environmentally mediated infections, such as vector-borne diseases, enteric pathogens and respiratory pathogens, which accounted for 40% of the global infectious disease burden in 2015 [10].

Infants and young children are especially vulnerable to unhealthy housing and crowding because they spend most of their time at home [11]. Housing interventions that reduce environmental infections are crucial for protecting their health. Pharmaceutical treatments often do not address environmental pathogen reservoirs and thus may not be able to achieve sustained disease transmission reductions on their own [10]. While decades of research have investigated specific housing-related interventions, including: water, sanitation, and hygiene (WASH), household air pollution, and bednets [12–21], other types of housing interventions have been relatively unexplored. Less prominent housing interventions designed to reduce environmentally mediated infectious diseases in low- and middle-income countries can be broadly categorized into two main areas: finished floors and housing upgrades for vector-borne disease prevention (Fig 1).

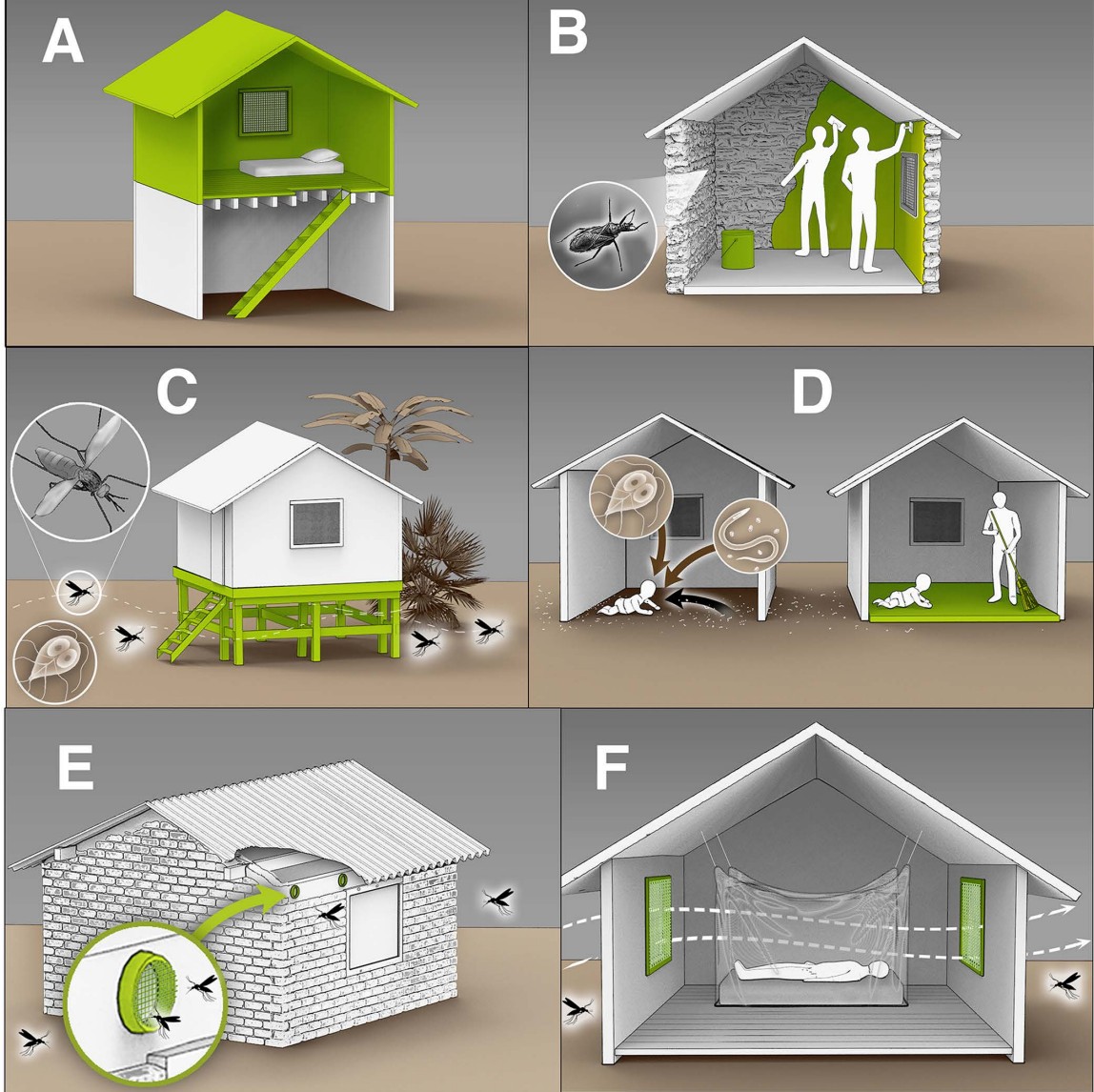

**Fig 1. Overview of housing interventions aiming to reduce environmentally mediated infectious diseases.** A) Constructing a second floor with a raised sleeping area to ensure airflow and decrease mosquito entry [24]. B) Applying wall plasters to smooth out crevices and prevent vector-borne diseases (e.g. Chagas) [45]. C) Lifting houses above the ground to reduce the risk of enteric and soil-transmitted infections and minimize mosquito entry [28]. D) Replacing dirt floors with finished floors to reduce exposure to microbial pathogens, including protozoa and soil-transmitted helminths [22,46]. E) Installing eave tubes to reduce mosquito entry while maintaining airflow [47]. F) Installing screened openings to reduce mosquito entry while ensuring sufficient airflow [24]. (Artwork by Victor Oleg Leshyk).

A number of observational studies have investigated impacts of replacing dirt floors with finished floors (e.g., wood, cement, or tile) on health. A study that replaced dirt floors with concrete in Mexico in the early 2000s (Piso Firme) found decreased parasitic infections, diarrhea, anemia, and adult depression, and improved children's cognitive development, adult quality of life, and adult happiness [22]. A 2023 meta-analysis reported that living in a home with improved floors was associated with 25% lower risk of any enteric or parasitic infection compared to living in a home with unimproved floors

[23]. Because all prior studies on this topic have been observational, the authors emphasized the urgent need for rigorous experimental studies [23].

A relatively large number of studies have also investigated housing improvements that reduce vector entry into the home in order to reduce malaria [24]. Nighttime can be a critical time for protecting household members from vectors [25,26]. Structural aspects of interventions include but are not limited to, lightweight and durable roofs with partially closed eaves, screened façades and openings, outdoor latrines, raised sleeping areas, and raised homes [24]. Additionally, housing upgrades to increase ventilation and airflow can reduce indoor carbon dioxide, which attracts mosquitoes [27], and lower indoor temperatures that may in turn increase the use of bed nets [28]. Two meta-analyses from 2021 and 2023 both found that housing interventions offer a significant protection against malaria and arbovirus infections [29,30]. The authors noted, however, that most studies were observational and the evidence base was weak [29,30].

The objective of this study was to explore facilitators of and barriers to funding, implementing, and scaling up housing improvements as health interventions to reduce environmentally mediated infectious diseases in low- and middle-income countries. Specifically, we aimed to 1) analyze the perceived relationship between housing and health among key informants, 2) identify barriers and success factors when implementing housing interventions, 3) describe the funding landscape of housing interventions for health, and 4) describe how successful housing interventions could be scaled-up. We employed a qualitative approach based on in-depth interviews with key informants who had direct experience implementing or working on housing interventions and environmentally mediated infectious diseases in low- and middle-income countries.

## Methods

### Ethics statement

This study was approved by the IRB (Protocol # 2023-11-16941) by the Office for Protection of Human Subjects (OPHS) of the University of California, Berkeley. In addition, the study was approved by the ETH Zurich Ethics Commission (EK-2023-N-317). Participants provided informed consent prior to each respective interview.

### Recruitment

We selected key informants (KIs) via purposive sampling, including 1) individuals in academic fields involved in researching housing interventions, 2) individuals involved in the development and passing of housing policies, and 3) individuals from non-governmental organizations working on housing improvements. We identified potential interviewees working in urban, peri-urban, and rural contexts through the authors' professional network, web searches, and snowballing technique. Fifty individuals were contacted via e-mail between 26.01.2024 and 13.03.2024, 28 replied, and 16 completed interviews (Table 1). There were no clear trends or patterns for those who chose not to participate. Participants provided informed consent prior to each respective interview.

### Data collection

We conducted interviews via Zoom videoconference that lasted between 18 and 70 minutes, averaging 40 minutes. Based on the participant's roles, two distinct semi-structured interview guides were used, tailored to capture insights from either a researcher/implementer or a policy perspective (S1 Text). Zoom was used to generate initial transcripts, which were subsequently checked against the audio files for accuracy.

### Data analysis

Following transcription, the interview transcripts were uploaded into the qualitative data analysis software NVivo (Version 14). Thematic analysis with an inductive, reflexive, experiential approach was applied to explore diverse experiences and

Global Public Health

**Table 1. Basic demographic information about key informants. Identifying information has been removed to protect key informants' anonymity.**

| Key informant | Current position | Experience relevant to housing interventions | Primary location of experience | Description | Interview guide used |
|---|---|---|---|---|---|
| 1 | Professor of Public Health in Asia | Developing housing innovations aimed at managing and eradicating various diseases such as malaria, respiratory tract infections, and enteric diseases | Tanzania and other sub-Saharan African countries | Researcher | Researcher/ Implementer |
| 2 | Professor of Public Health Entomology in Europe | Modifying modern housing with increased ventilation, closed eaves, and screenings | Gambia | Researcher | Researcher/ Implementer |
| 3 | Head of Architecture in a European university | Developing housing innovations aimed at managing and eradicating various diseases such as malaria, respiratory tract infections, and enteric diseases | Tanzania and other sub-Saharan African countries | Researcher | Researcher/ Implementer |
| 4 | Researcher in an academic organization in Africa | Improving housing temporary structures for migratory farmers to reduce vector-borne diseases | Tanzania | Researcher | Researcher/ Implementer |
| 5 | Researcher in an academic organization in Africa | Modifying eaves and screening of windows and doors | Sub-Saharan Africa | Researcher | Researcher/ Implementer |
| 6 | Professor of Environmental Design and Engineering in Europe | Researching the relationship between housing and moisture-related diseases | Global | Researcher | Researcher/ Implementer |
| 7 | Researcher in a European University | Creating and evaluating housing improvements to reduce heat- and cold exposure, respiratory infections, and indoor mosquito density | Global | Researcher | Researcher/ Implementer |
| 8 | Senior leadership role at a social enterprise in Africa | Delivering sustainable flooring solutions and novel housing solutions in developing countries | Rwanda and Uganda | Practitioner/ Service manager | Researcher/ Implementer |
| 9 | Senior leadership position at an international non-profit organization | Implementing sustainable solutions, such as concrete floors, to vulnerable populations | Mexico and Central America | Practitioner/ Service manager | Researcher/ Implementer |
| 10 | Senior leadership position at an international NGO | Overseeing implementation of housing interventions to combat several diseases | Global | Service manager | Researcher/ Implementer |
| 11 | Country Program Director in an international non-profit organization | Assisting the implementation of housing improvements for vulnerable populations; assistance in the improvement of housing policies | Lesotho | Service manager/ Policy advisor | Policy |
| 12 | Senior Urban Specialist at an international development bank | Working on interventions to increase access to land through land use planning; projects that finance housing improvements; supporting governments on developing housing policies | Mexico and Central and South America | Policy advisor | Policy |
| 13 | Senior Urban Specialist at an international development bank | Supporting client LMI-countries to improve existing housing & reduce vulnerability to natural hazards & climate change through financial | Southeast Asia | Policy advisor | Policy |
| 14 | Country Program Director in an international non-profit organization | Specializing in disaster-resilient housing solutions in vulnerable regions | South America | Policy advisor | Policy |
| 15 | Country Program Director in an international non-profit organization | Specializing in disaster-resilient housing solutions in vulnerable regions | South America | Policy advisor | Policy |
| 16 | Senior leadership position at an international non-profit organization | Overseeing policy-relevant evidence generation and impact evaluation of housing interventions | Global | Policy advisor | Policy |

perspectives and identify themes from the data without preconceived categories [31,32]. Our approach was rooted in reflexive thematic analysis [33], which emphasizes identifying patterns and significant insights and prioritized the depth and relevance of themes over counts (e.g., content analysis). The analysis followed a recursive process comprising six overlapping phases: 1) familiarization with the data; 2) systematic generation of open codes which identified and briefly described relevant and interesting segments; 3) generation of initial themes by collating relevant codes; 4) further development and revision of themes 5) final definition and naming of themes; 6) writing the report, incorporating illustrative examples and interpretations.

## Results

### Themes

Based on the in-depth interviews, seven themes were identified in the analysis of stakeholder's perspectives on housing interventions. These included: 1) Communicating the multifaceted impacts of housing interventions, 2) Breaking silos and sector-building, 3) Building the evidence on health impacts and cost-effectiveness, 4) The role of governments, private institutions, and communities in scale-up and need for collaboration, 5) Financing challenges and priorities, 6) Policy and regulatory framework, 7) Addressing contextual and operational challenges.

### Communicating the multifaceted impacts of housing interventions

Multiple KIs emphasized the wide-ranging effects of housing interventions on health, noting that they extend beyond addressing one single disease. These included vector-borne diseases like malaria and dengue fever, diarrheal diseases, respiratory illnesses, and water-borne diseases like cholera and typhoid fever.

KI 2 described how the combination of factors, including recurrent attacks of malaria, diarrhea, and poor nutrition, collectively diminish a child's resilience in a low-income setting, ultimately increasing risk of mortality.

*"So, although the final death is due to malaria, if that child was fit and healthy, they'd be more resistant, less likely to die from malaria. And that's true for all the major killer diseases. It's the fact that the child has everything else since [they] were born. [The child] had a whole succession. Because [he or she] lives in a dirty environment."* (KI 2, Researcher)

On a similar note, KI 6 highlighted how numerous factors of the built environment can influence the residents' well-being, including but not limited to temperature, moisture, overheating, lighting, acoustics, and overcrowding.

While discussing the health implications of housing interventions, several KIs emphasized their significance in preventing diseases.

*"We call healthy housing an architectural vaccine. That's the way to really do a lot of disease prevention."* (KI 8, Practitioner/Service manager)

Multiple interviewees indicated that by enhancing housing conditions, residents experience fewer illnesses, enabling them to pursue their professional, social, and personal goals more effectively. Describing the impacts of replacing dirt floors with finished floors, KI 9 noted:

*"We have been able to see people, when you take away one significant worry from their life, like housing, they are able to expend the mental, and spiritual, and physiological energies towards something else, like improving their education, going after a better job, even just child rearing – taking care of their children. The burden on that becomes slightly lessened because one of their major needs is now fixed and taken care of."* (KI 9, Practitioner/ Service manager)

*"Prevention is always the poor bedfellow of health interventions. People usually want nice hospitals and drugs at the local doctors; and of course that's understandable. But so many other problems could be prevented. And so, it's the importance about prevention."* (KI 2, Researcher)

Several KIs expressed the view that housing has multi-dimensional impacts, encompassing economic, social, and environmental dimensions, and that housing is linked to several SDGs.

*"Housing is one of those things that touches everything. It touches job, it touches climate and touches health. It touches everything."* (KI 8, Practitioner/Service manager)

*"When you look at it [the SDGs], and you go through the list asking, 'Can I achieve that without good housing?', then the answer is no."* (KI 13, Policy advisor)

Many KIs highlighted the profound mental health impacts of housing, emphasizing factors such as decreased stress levels and depression. KI 13 described how mental health problems are more challenging to cope with in a polluted, noisy environment. Additionally, several KIs indicated that factors like social status, comfort, and pride associated with one's home affect mental health.

KIs highlighted numerous environmental factors associated with housing, including the need to construct homes with sustainable materials instead of high-carbon-footprint materials such as concrete. KI 3 mentioned that the high deforestation in India led to a new law prohibiting the use of wood in buildings. Similarly, KI 11 described how wood is still being used as a traditional fuel source in Lesotho for cooking and warming houses, which is causing harm to the environment.

*"It's really obvious that over-dependence on these trees has a negative impact on our environment because they are being over-exploited."* (KI 11, Service manager/Policy advisor)

Some KIs indicated that growing concerns about climate change have elevated attention toward housing interventions, particularly climate-resilient housing. KI 13 commented that the impacts of climate change, including the higher frequency of natural disasters, floods, and heavy winds, amplify awareness of policy makers regarding the interconnectedness between housing and health. On a similar note, KI 14 observed that decision-makers tend to pay more attention to housing when risks become more palpable. Even countries not directly impacted by a natural disaster, as described by KI 15, are spurred to action to improve housing by global disasters like earthquakes.

Multiple KIs underscored the persistent need for heightened awareness of housing, which receives less attention than other topics, such as climate change. KIs deliberated on effective strategies to disseminate the importance of their work and the critical role of housing in addressing health disparities.

*"I feel very strongly that the dissemination, or the communication effort of what we do is almost as important as the implementation work. Because if we get the message out to more people how to do this work, […] I think we can start to get more impactful benefit for more people."* (KI 10, Service manager)

Multiple KIs pointed out to the significant opportunity within the housing sector due to anticipated population growth in coming years. They highlighted the potential to build sustainable houses from the ground up.

*"For example, in sub-Saharan Africa, the population will increase by 1.2 billion people. So, that's like the current population of China or India adding to the continent. They will build houses. Whatever you do. And you can build efficient houses off from scratch, which is great."* (KI 2, Researcher)

**Breaking silos and sector-building**

Several KIs highlighted a prevailing lack of recognition among individuals and sectors regarding the critical link between housing and health. For example, KI 3 pointed out that malaria research has predominantly been driven by biologists, who have typically neglected the significance of housing.

> *"I don't think many health professionals are kind of thinking in that way, how the urban fabric for instance is really important for the health of the population."* (KI 3, Researcher)

KI 13 illustrated how ministers of education often do not see the connection between children's educational success and their living conditions. Moreover, KI 13 stated a similar challenge in the case of the ministry of health, saying:

> *"You know, you can have a lot of gastro-intestinal diseases, respiratory diseases. And all takes place at the home because the house has mold, or you have a dirt floor, or you don't have toilets. Or you know, the chickens are walking around your house where your kids play."* (KI 13, Policy advisor)

On a similar note, KI 14 suggested that architects and engineers should be educated differently, considering health factors when building a house. Likewise, KI 2 noted that architects, engineers, and city planners may possess a limited understanding of mosquitoes and the diseases they transmit and underscored the necessity for health considerations to transcend the boundaries of health departments, which traditionally addressed such matters in isolation:

> *"This is a multi-sectoral response that's required. […] What we've got to do here is break across silos. There are huge opportunities for changing things."* (KI 2, Researcher)

KI 7 articulated that a challenge for housing is that it does not fit squarely within a specific sector, like energy or health.

> *"I think that's the issue, that this kind of work doesn't sit anywhere and there's no clear place for it. So that sector-building work needs to be done."* (KI 7, Researcher)

**Building the evidence base on health impacts and cost-effectiveness**

Numerous KIs underscored the ongoing need for further evidence on the impacts of housing interventions on health, both for securing funding and effectively scaling up housing improvements. Talking about the scarcity of direct evidence of health impacts, KI 6 noted that research needs to focus on:

> *"Understanding in which ways buildings affect health, but it's also about developing the methodology to assess. Because how do I know that this room for example is affecting me? So, it's trying to understand the criteria to assess buildings."* (KI 6, Researcher)

KIs consistently emphasized the critical role of a comprehensive cost-benefit analysis; for example, KI 5 noted that cost-benefit analyses are needed to determine which housing modifications are most effective for specific diseases.

Multiple respondents indicated multiple factors that should be considered when measuring cost-effectiveness. For example, several KIs noted that households would have fewer repairs (i.e. lowering operation and maintenance costs) if construction was done with more durable materials.

*"So, by creating a better home, which you can have for the next 30 years, you increase the welfare of people. And I mean, that also has to be kind of factored in if you want to assess the benefits of new housing."* (KI 1, Researcher)

Moreover, KI 1 highlighted the importance of successfully showing housing's impacts on multiple health aspects to justify the cost of housing interventions. KI 4 pointed out that housing improvements often benefit all household members, and KI 2 stressed that housing interventions can protect people for decades; both of these factors will generally increase their cost-effectiveness relative to individual-based interventions.

KI 14 emphasized that by improving housing conditions, expenses on other things, such as treatment and medications would likely decrease.

*"So, if you can have a secure housing, or a core housing, then even your payments in health are going to be reduced, because you are not going to get sick as often."* (KI 14, Policy advisor)

Similarly, KI 5 noted, that:

*"If you look at the cost [of the screening intervention] versus the cost that a household would also spend in managing malaria, it is still cheaper […] It's just the lower cost doing the modification then spending $75 for example in a year to treat malaria – if the children don't die anyway. Because there's also the element of death. If you factor in death that you could also suffer as a family as a real cost of not doing the modification."* (KI 5, Researcher)

KI 10 acknowledged that housing interventions can be extremely expensive and not as cost-effective as certain pharmaceutical interventions.

*"Even when we're talking about cost-effective household improvements, you can't get down, as you're not talking about a dollar vaccine kind of thing. You're really at a higher rate and so even if they're longer-term, it just ends up being an expensive intervention and therefore a proper study ends up being a bulky budget."* (KI 10, Service manager)

In discussing its role in the policy-decision process, KI 16 emphasized the importance of evidence in the policy making process, even if it is not the only relevant factor, saying:

*"I think these studies [on housing and health] have definitely helped put in-situ upgrading on the radar of governments, nonprofits, multilaterals, as interventions that may do more to improve human welfare than just improving living conditions. That these can actually have positive effects on human development indicators."* (KI 16, Policy advisor)

*The role of government, private institutions, and communities in scale-up and need for collaboration*

A recurring question became whether a top-down or a bottom-up approach is most effective in scaling up housing improvements. Several KIs underscored the critical role of government involvement in effectively scaling up housing interventions.

*"Housing programs need to have the national government, the federal government behind. Because it's only the government level that has the fiscal instruments to do it."* (KI 12, Policy advisor)

*"We think that the only way this is scalable is if we work with governments, because they are the ones that are mandated by the constitution to provide dignified housing for all."* (KI 15, Policy advisor)

KI 16 highlighted that in the case of the Piso Firme project, the initial drive for the study evaluation came from the Mexican government, which was interested in generating evidence for Piso Firme as a potential low-cost scalable model [22].

*"These were high-level Mexican policy makers looking at the evidence, looking at the cost-effectiveness and making policy decisions on where to allocate their social budget for housing. […] Piso Firme was able to scale because there was a large government behind it."* (KI 16, Policy advisor)

KI 3 expressed a critical view regarding extensive government involvement in the scale-up approach, stating:

*"I would not be too keen on going with the big governmental institutions. […] I don't think it's solved by institution. I think, a lot of it is solved by private investment, combined with good planning and good support from the government. […] But when the government starts to build, or big institutions start to build, it always goes wrong. It needs to be something in between."* (KI 3, Researcher)

KI 9 outlined a strategy focused on prioritizing assistance to those in need and emphasized the significance of fostering community connections to achieve sustainable long-term and replicable outcomes.

*"It wasn't necessarily about institutions. We obviously want to tackle the institutional disparities, but it wasn't about starting at the top. It was more about starting at the grassroots, where we can find people who are connected to those larger institutions but who are the ones doing the work, who are the ones that are going to be able to actually implement these processes."* (KI 9, Practitioner/Service manager)

Several KIs discussed how a government's political orientation influences its housing agenda and resource distribution. KI 12 delineated how a transition from a right-wing to a left-wing government precipitated a change in the nation's housing policy. Similarly, KI 13 noted:

*"So, when you have a transition […] of government […] the allocation of housing and the type of work you do becomes completely different."* (KI 13, Policy advisor)

Considering the dynamic nature of political landscapes and the brevity of legislative tenures, KI 2 underscored the need for strategies resilient to political fluctuations.

*"So, what we're talking about is long-term strategies that are supported irrespective of the political color of local politicians."* (KI 2, Researcher)

Several KIs recognized the significant influence of the private sector in driving the direction of housing scale-up. For instance, KI 15 emphasized the pivotal role of the chamber of commerce in the construction sector as a major policy driver in Colombia. On a similar note, KI 12 observed:

*"I mean the private sector has a very important role. Of course you need to be careful. […] That you [don't] have housing policies that are led by the private sector completely, that only serve their interest. It happens quite often in many countries."* (KI 12, Policy advisor).

The importance of community involvement, as emphasized by several KIs, lies in its role in facilitating the understanding of community perspectives on housing conditions, securing willingness to implement housing interventions, and establishing project-based policies.

*"You need to get the community involved, the people whose lives you are trying to improve. To ask them about and explain [to] them what the issues are and to see their initiatives for improving it."* (KI 2, Researcher)

Several KIs delineated the consequences of insufficient community involvement in health interventions, noting disparities in perceived priorities between communities and other stakeholders. KIs provided examples of when health interventions have been repurposed by communities for more immediate priorities, such as using bed nets for fishing (KI 5), safeguarding poultry, and protecting crops from insects (KI 6).

In discussing the disparity between public stakeholders or government finance sectors and the communities, KI 7 emphasized that while governments tend to prioritize disease indicators, communities often have numerous other housing-related considerations that hold greater importance to them.

*"What the biggest burden of disease is […] doesn't capture the lived experience of somebody living in a poor-quality house which might have many more impacts in terms of the house. The time and energy it takes to deal with a problematic […] house is not considered."* (KI 7, Researcher)

Numerous KIs stressed the importance of extensive stakeholder collaboration and local capacity building to ensure the effective scale-up of housing improvements.

*"It's not only like, 'okay, because we already changed the policy, things are going to happen just like that'. You also need to have like that cycle and that pipelining, which you can also change, or improve technical competencies at the local level. So, I think our challenge is how can you make a pipeline in order to transfer knowledge [about how to build correctly] to all parts of the value chain. And also, how can we learn from it."* (KI 14, Policy advisor)

*"And then we need to have people all the way down to like the grassroots level and people who are really doing implementation on the ground, and we need to have like everybody in between. We need to have researchers, we need to have practitioners, everybody involved so that I think we can really make some progress on this."* (KI 10, Service manager)

KI 13 underscored the potential risk of uneven stakeholder influence, emphasizing the imperative of guiding stakeholders toward a unified mission to ensure collective progress.

### Financing challenges and priorities

While KIs noted diverse financing options for housing interventions, including government funding, mortgages, and private sector investment, they consistently highlighted inadequate financing systems and funding as a major obstacle to implementing and scaling up housing interventions. KIs noted that there is a preference to fund disease-specific, pharmaceutical interventions over broader housing interventions, possibly due to limited evidence on housing interventions and higher cost.

KI 10 noted a common response from potential funders, expressing skepticism about housing as a proven health intervention. A dilemma arising from the scarcity of existing evidence and lack of interest in supporting the generation of new evidence was described by KI 7:

*"I think it's a bit chicken and egg, like there's no data on it but then there's nobody collecting the data that we need and therefore it's not getting the exposure, or the focus."* (KI 7, Researcher)

KI 9 highlighted a tension in the funding landscape, noting that smaller funders often prioritize seeing direct impact over rigorous research. In comparison, bigger funders demand more stringent evidence evaluation but do not prioritize and focus on housing.

Several KIs mentioned that housing is competing for budget and resources with other sectors and stakeholders, posing another challenge in financing housing improvements. Discussing the reluctance of some governments to invest in housing interventions, KI 12 noted that this is:

*"Either because they don't want to raise taxes, or they don't want to cut expenditures in other sectors to just spend in modern housing."* (KI 12, Policy advisor)

Similarly, KI 9 noted that housing sometimes is:

*"Competing for space, competing for solutions between NGOs and between those who are actually trying to do good work."* (KI 9, Practitioner/Service manager)

## Policy and regulatory framework

Multiple KIs highlighted the pivotal role of policy in enabling the sustainable and efficient scaling up of housing interventions, as they dictate the manner of construction while also regulating and standardizing the process. According to KIs, promoting social justice and human rights within housing policies and constitutions is another cornerstone of the regulatory framework. Using an example involving screening windows and doors, KI 2 discussed the role of policies in housing interventions:

*"You could shape the buildings of the future by having local bylaws, or enacting bylaws which say that all houses should be screened. You could reduce tax on importation of screening [to]encourage house screening."* (KI 2, Researcher)

*"So, if you could make local zoning rules and building codes that are actually reasonable, that would be good."* (KI 3, Researcher)

KI 12 stressed that the existence of housing policies does not guarantee improved housing conditions.

*"We're seeing a lot of countries that have policies and no implementation because it is just like a piece of paper."* (KI 12, Policy advisor)

Several KIs highlighted that poorly or inadequately formulated policies can impede the delivery of healthy housing solutions. They observed that housing policies in several countries contain blind spots, for instance because they tend to over-emphasize economic growth, and that they are often disconnected from the communities or those implementing housing improvements. KI 13 stated:

*"If I am a minister of housing that focuses on economic growth, I can be very convincing and I can structure all my products around that. But there is a huge price for the vulnerable families."* (KI 13, Policy advisor)

*"I think there's a real gap between people who are writing policy and people who are implementing the work and getting action done."* (KI 10, Service manager)

Another aspect discussed by KIs is the issue of housing ownership, which adds to the complexity of the regulatory framework. KI 7 described that, unlike the public nature of the healthcare system, housing operates within a complex private market with various ownership models, posing difficulties to widespread interventions. Discussing the likelihood of housing interventions being scaled up, KI 7 noted:

*"In a situation where you have public ownership of housing and social housing, and you have more involvement from the state, then it's probably more likely. But when you have very privatized markets, where land is really expensive, when housing itself is a commodity, […] then it becomes quite difficult."* (KI 7, Researcher)

**Addressing contextual and operational challenges**

Multiple respondents pointed out that a lack of skills and knowledge among local populations hinders the implementation and scaling-up of housing improvements. This can result in poor quality of construction and unsafe buildings. KI 8 noted:

*"The labor market doesn't exist really to very easily train a bunch of masons. So, we need a whole apprenticeship program."* (KI 8, Practitioner/Service manager)

In discussions on operational challenges, KIs emphasized considerable logistics and costs of transporting construction materials to some areas. Additionally, they noted the lack of supply chain and distribution networks as further obstacles.

*"Then another market failure is on supply chain. […] there's no FedEx here, there's not like a UPS. Like, every person has to create their own last-mile distribution network. And that is very expensive."* (KI 8, Practitioner/Service manager)

KI 3 highlighted the low degree of industrial manufacturing in some countries and how the need for manual labor to build homes by hand dramatically raises costs and reduces the scaling up of improved housing.

*"I mean, the success of a house in our parts of the world [high income settings] is because it's highly industrial today and you can build very efficient and also very good."* (KI 3, Researcher)

Multiple interviewees emphasized the necessity of considering contextual factors specific to each region or community when implementing or scaling up housing policies and interventions, as the materials, climate, and cultural norms and practices vary across different geographical areas.

*"So, what kind of environment? What kind of people? How do they live their lives? You, know, it's important because you will not take the same interventions everywhere."* (KI 5, Researcher)

*"I would go with that context-specific approach where I would say, before allocating resources, I would do my groundwork and say, 'which is the biggest challenge in this particular region?'"* (KI 4, Researcher)

According to KI 9, a consequence of not correctly adjusting programs to their context is that solutions and interventions have become very templated, where people rely on what worked in the past and try to apply it in the same manner in another context.

*"Just because it's the same problem doesn't necessarily mean that it necessitates the same solution."* (KI 9, Practitioner/Service manager)

## Discussion

This study contributes to the literature on housing and health by exploring facilitators of and barriers to funding, implementing, and scaling up housing interventions as health interventions. We identified seven major themes from key informant interviews that shed light on the multifaceted nature of housing interventions and their implications for public health.

KIs unanimously agreed that housing modifications could reduce transmission of environmentally mediated infections as well as a broader set of outcomes, including mental health and life satisfaction. This finding is in accordance with previous literature showing reduced depression rates and stress levels upon housing improvements [22]. Similarly, two studies conducted in Ecuador [34] and Vietnam [35] showed that housing satisfaction is a strong predictor and key driver of life satisfaction, underscoring the importance of housing interventions. KIs consistently stated that housing interventions should be prioritized within health and development agendas as a complement to existing strategies, such as WASH interventions.

KIs emphasized a pervasive shortage of funding and financial systems for implementation and scale-up of housing interventions. A 2019 study elucidated that broader measures like housing interventions are usually less recognized than disease-specific interventions and, consequently, lack financial support [27]. As several KIs indicated, promoting awareness of the significance of housing interventions is essential to garner greater support and investment. KIs articulated a need for more rigorous evidence on health impacts of housing as well as precise cost-benefit analyses, including a wide range of outcomes, to attracting donor funding [27]. However, they also noted a lack of interest in and funding to generate new evidence on housing interventions. Given the need for substantial investment to complete housing upgrades, housing programs may be more successful when aligned with other programs and campaigns to target broader benefits, beyond health.

With regard to implementation and scale-up of housing interventions, KIs also revealed that community involvement and governmental commitment are key factors. A 2012 qualitative study on barriers to scaling up health interventions in low- and middle-income countries found that initiatives often fail without engaging local implementers and recipient communities [36]. Moreover, the author emphasized the crucial role of national leaders in scaling up, a standpoint echoed by multiple KIs who acknowledged the pivotal role of government involvement in scaling up [36]. There are likely many important examples that could be used for effectively scaling up housing interventions. A notable one is the Swachh Bharat Mission (Gramin), or 'Clean India Mission' program that aimed to increase sanitation coverage in rural India, which is estimated to have shifted rural sanitation coverage from 39% to nearly 95% between 2014 to 2019 [37]. Research suggests that the Swachh Bharat Mission, which cost approximately $9.7 billion USD [38], succeeded in-part through committing to ambitious targets, utilizing more modern approaches to messaging, marketing and monitoring, and publicly rewarding and recognizing implementers involved in achieving the programmatic goals [37]. Given the potentially high capital costs for some housing improvements, a combination of grants, loans and tariffs, similar to Swachh Bharat, would likely be required to achieve the level of funding needed for housing improvements at scale. A key takeaway from many other national programs that have been taken to scale is that governments are best positioned to coordinate the implementation of housing programs that will need to be designed to reach the poorest households [39,40].

KIs stressed the urgent need to break silos and collaborate across disciplines to stimulate research and action on housing and health. The importance of strong intersectoral collaboration has been emphasized for other interventions; for example, the success of Singapore's dengue control program was attributed to close cooperation among various government ministries, town councils, communities, the private sector, and academic and research institutions alongside a legislative framework [41]. Similarly, KIs emphasized the fundamental role of project-based and context-specific housing policies in addressing social determinants of health such as housing. This sentiment resonates with a 2021 comparative housing policy study, emphasizing the importance of integrated policy responses across relevant areas to effectively address the strong association between housing and health [42]. Climate change will further exacerbate the need for improved housing [43] and as temperatures increase, the need for housing with better insulation, ventilation, and cooling mechanisms to protect household members will rise. Many health outcomes associated with high temperatures in homes have already been observed, including heat stroke, respiratory and cardiovascular hospitalizations and deaths [44].

A limitation of this study is that we were not able to interview any government-based key informants even though we invited them. We were able to interview only one individual who worked closely with the Mexican government during the

Piso Firme program. The perspectives of government employees would have shed light on potential barriers to funding, subsidizing, and zoning and permitting processes for housing interventions to improve health. Multiple KIs mentioned that governments would play a critical role in scaling up housing interventions, in part due to their financial resources but also through constitutional mandates in some countries to provide housing for all. Non-response from government-based key informants may reflect a lack of focus on housing and health interventions – besides ones made for sanitation, household energy or bed nets – in government programs. Additionally, KIs remarked that governments may face difficult trade-offs when funding housing interventions, such as choosing between offering tax cuts or reducing spending in other areas.

There were several other limitations of this study. The sample size was limited by the number of key informants we could identify who have worked on housing interventions aimed at reducing infectious diseases; the small number of stakeholders may have restricted the viewpoints and themes identified in the analysis. For example, certain barriers such as lack of land ownership or the limitations faced by renters, as well as household crowding were not raised in the interviews. Furthermore, we did not have KIs who represented communities; these stakeholders could have provided valuable insights. Engaging community residents involved in housing improvement programs ended up being beyond the scope of this study. Similarly, we only invited KIs who could complete the interview in English, which further limits the diversity of perspectives in this study. This study did not address the contribution of environmentally related infections outside the home, despite many older children and adults spending much of their time outside the home. Lastly, the broad spectrum of housing interventions was discussed in a generic manner, which overlooks nuances specific to particular interventions.

## Conclusions

In conclusion, this study underscores the potential role of housing interventions in mitigating environmentally mediated diseases, which can address multiple health challenges simultaneously. Recommendations highlighted a need to foster collaboration across diverse sectors and stakeholders, as well as a need to raise awareness of the importance of housing interventions. Working across disciplines and sectors will be imperative to improve housing and achieve sustainable development goals. Furthermore, efforts to prioritize housing interventions on the policy agenda and to align interests among diverse stakeholders are essential for their effective implementation and scale-up. Additionally, investments in generating rigorous evidence and promoting contextually relevant policies will be important for advancing the efficacy and impact of housing interventions in improving public health outcomes.

## Supporting information

**S1 Text.  Semi-structured interview guide.**
(PDF)

**S1 Checklist.  Inclusivity in global research checklist.**
(DOCX)

## Acknowledgments

Research reported in this publication was supported in part by the National Institute of Child Health and Human Development under Award Numbers R01HD108196 (PI: Benjamin-Chung). The content is solely the responsibility of the authors and does not necessarily represent the official views of the National Institutes of Health. Jade Benjamin-Chung is a Chan Zuckerberg Biohub Investigator.

## Author contributions

**Conceptualization:** Matthias Acklin, Jay P. Graham, Jade Benjamin-Chung.

**Data curation:** Matthias Acklin.

**Formal analysis:** Matthias Acklin.

**Investigation:** Matthias Acklin, Jade Benjamin-Chung.

**Methodology:** Matthias Acklin.

**Project administration:** Matthias Acklin.

**Supervision:** Jay P. Graham, Jade Benjamin-Chung.

**Writing – original draft:** Matthias Acklin.

**Writing – review & editing:** Matthias Acklin, Jay P. Graham, Jade Benjamin-Chung.

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
