## [Decision Letter · Decision Letter 0]

9 Dec 2024

PGPH-D-24-02175

From the Ground Up: Stakeholder Perspectives on Housing Interventions to Reduce Environmentally Mediated Infections

Dear Dr. Benjamin-Chung,

Thank you for submitting your manuscript to PLOS Global Public Health. After careful consideration, we feel that it has merit but does not fully meet PLOS Global Public Health’s publication criteria as it currently stands. Therefore, we invite you to submit a revised version of the manuscript that addresses the points raised during the review process.

Please provide a point by point response to the reviewer comments.

We look forward to receiving your revised manuscript.

Kind regards,

Joanna Tindall, PhD

Staff Editor

Journal Requirements:

2. Please provide separate figure files in .tif or .eps format.

For more information about figure files please see our guidelines:https://journals.plos.org/globalpublichealth/s/figures 

Additional Editor Comments (if provided):

Reviewers' comments:

Reviewer's Responses to Questions

**Comments to the Author**

1. Does this manuscript meet PLOS Global Public Health’s publication criteria? Is the manuscript technically sound, and do the data support the conclusions? The manuscript must describe methodologically and ethically rigorous research with conclusions that are appropriately drawn based on the data presented.

Reviewer #1: Partly

Reviewer #2: Yes

Reviewer #3: Yes

2. Has the statistical analysis been performed appropriately and rigorously?

Reviewer #1: N/A

Reviewer #2: N/A

Reviewer #3: N/A

3. Have the authors made all data underlying the findings in their manuscript fully available (please refer to the Data Availability Statement at the start of the manuscript PDF file)?

Reviewer #1: Yes

Reviewer #2: No

Reviewer #3: Yes

4. Is the manuscript presented in an intelligible fashion and written in standard English?

Reviewer #1: Yes

Reviewer #2: Yes

Reviewer #3: Yes

5. Review Comments to the Author

Reviewer #1: Thank you for the opportunity to review this article.

Housing can support or undermine the health of householders from a range of aspects. This research contributes to the documentation of the areas of vulnerability and/or protection for human health from their housing conditions.

While the topic area is strong, this study has several flaws that do not make it a robust piece of research. My detailed comments are below.

Abstract:

- this does not set up the paper very well. It cites an example of housing improvements as being 'finished floors', which is not the example I was expecting (such as running water, heating and cooling, crowding levels etc).

- the results summary is very broad when details would have been helpful.

- 'infections' are mentioned broadly, but details (eg infection names or groups) would have been beneficial.

- overall, the abstract lacks details about the study- which is then reflected in the body of the paper.

Background:

- the scope of the research is extremely broad; there is insufficient detail on the countries / regions of focus and on the issues of concern (eg transmission of infections); there is a focus on finished floors and vector borne infections but these two areas do not have a logical link to each other; climate change issues are raised without sufficient connection to housing... in sum, the scope is extremely broad and prevents tangible results being identified that can then be implemented into housing developments of policy.

Methods:

- the sample size (16 key informants) was very small. This can be justified, though, if the authors wish to do so.

- the key informants are said to describe low- and middle-income countries. Can more detail please be provided regarding the countries about which or from which the informants were reporting? This may enable reviewers to understand whether this is a strong sample from which findings can be assessed as robust.

Discussion:

- some literature is referred to, but it would be better to return to this literature once it has been cited earlier (eg in the Background).

I retain several queries:

- I am intrigued that crowding was only mentioned once but not extrapolated. Crowding has so many impacts on human health in housing settings that this is a key aspect to consider.

- I was hoping that the authors may propose a way to evaluate how health is impacted by housing and housing improvements. It was a shame that this was not raised.

Reviewer #2: The authors did not make the data publicly available as per PLOS data policy due to Identifiable content. The author declares that data used to generate the outcome will not be available to protect participants' privacy.

Reviewer #3: Thank you for the opportunity to review this well-executed and insightful manuscript. The study addresses an important and timely topic, with housing interventions holding significant potential for improving health outcomes and advancing Sustainable Development Goals in the face of population growth and climate change. The paper is well-organized, clearly written, and provides valuable perspectives on barriers and facilitators to implementing these interventions in low- and middle-income countries. Additionally, the thematic analysis is robust and offers meaningful insights. However, there are notable gaps in the perspectives included, particularly regarding the diversity of voices represented and the exploration of actionable takeaways. Strengthening these aspects could further enhance the study’s impact and utility for practitioners, policymakers, and researchers alike.

Major:

1. The manuscript’s title emphasizes "stakeholder perspectives on housing interventions," yet the perspectives presented are predominantly from academics—who may have limited direct involvement in large-scale implementation—and practitioners or policymakers affiliated with large, often multinational organizations. Notably absent are the voices of local or national government officials, despite the frequent identification by respondents as government officials being critical stakeholders in the success or blocking of housing interventions. Similarly, perspectives from grassroots organizations or residents of substandard housing are not included. While the exclusion of residents and grassroots organizations is understandable given the scope of this global study, the absence of government perspectives represents a significant limitation. Clarification regarding whether government officials were contacted but declined to participate, or were intentionally excluded, would be valuable. Addressing this gap or providing a rationale for their absence would strengthen the paper’s comprehensiveness and applicability to real-world policy and implementation.

2. Two key areas were notably absent from the interviews or their inclusion in the paper, which I found surprising given their potential relevance to the study’s focus. First, the contribution of environmentally related infections outside the home was not addressed, despite the fact that most individuals, particularly in LMICs, spend significant time outside their residences. Exceptions such as young children or the elderly—important at-risk groups—could be acknowledged to better contextualize the risk landscape. Second, barriers to improving housing, including lack of land ownership or the challenges faced by renters, were not discussed. These are critical structural issues that often underpin the feasibility and sustainability of housing interventions and could enrich the discussion. The authors may wish to consider briefly addressing these points in the discussion section. Additionally, it would be helpful to know if there were other themes in the literature that the authors anticipated emerging in interviews but did not. A reflection on these gaps could provide valuable insights into the study’s scope and help situate its findings within broader conversations on housing and health.

3. The authors emphasize the need to gain traction for housing improvements as a strategy to enhance health outcomes. However, evidence from other public health initiatives, such as WASH, indicates that health alone may not be a compelling motivator for action by communities or governments. As noted in the manuscript, housing improvements generate a broad array of co-benefits beyond health, including economic stability, educational advancements, and environmental sustainability. This raises an important question: might advocacy for housing improvements benefit from aligning with campaigns targeting these broader benefits? For example, the introduction connects housing improvements to climate change and WASH interventions, but this link is not revisited in the discussion. Exploring lessons from these fields could strengthen the argument for housing interventions and provide valuable insights into potential advocacy strategies.

4. Additionally, parallels with WASH suggest that localized improvements (e.g., a few upgraded homes in a slum neighborhood) may not yield measurable health impacts due to continued exposure from the broader environment. Yet, widespread improvements may generate community-level benefits. If this is the case, it underscores the critical role of government in scaling interventions to achieve population-level impacts. The authors could consider discussing whether such collective benefits might strengthen the case for public investment in housing and whether collaboration with sectors like WASH and climate change could amplify advocacy and implementation efforts.

5. While the manuscript is very strong overall, the discussion and conclusion sections do not fully match the rigor and depth of the earlier sections. These could be enhanced by addressing the comments raised above and by providing more concrete recommendations on which stakeholders should lead action and the strategies they might employ. Offering actionable insights tailored to different stakeholder groups—such as policymakers, practitioners, and researchers (and ideally government and communities) —would provide a clearer pathway for translating the study’s findings into practice and future research. Additionally, the conclusion could be strengthened by revisiting and emphasizing the stakes involved in not improving housing, as articulated effectively in the introduction.

Minor

Line 117 – was there anything systematic about the non-response? Were any groups in the original sample under or not represented in the final interviews?

Line 154 – suggestion to replace the interviewee’s use of “it” to refer to children with bracketed [they] [he or she] or [the child].

Line 188 – suggestion to replace the “traditionally used materials” referring to concrete with another word --- “traditional” is used later to describe things like wood. Or if keeping the word traditional, the authors could make it clear which tradition/traditional for whom/as of when.

Line 265 – I believe “with” is a typo and should be replaced with “for”

Line 313- suggestion to add the reference for the Piso Firme project (it’s in the intro but readers may have forgotten what the program is)

Line 376 – it’s unclear to me who the “them” is referring to

Line 419 -_“tend to over-emphasized economic factors” feels vague – could you provide an example(s).

Line 511 - suggestion to remove “which may have” and just say “limits the diversity”

Line 513 - suggestion to remove “may have” and just say “overlooks nuances”

6. PLOS authors have the option to publish the peer review history of their article (what does this mean?). If published, this will include your full peer review and any attached files.

**Do you want your identity to be public for this peer review?** For information about this choice, including consent withdrawal, please see our Privacy Policy.

Reviewer #1: No

Reviewer #2: No

Reviewer #3: **Yes: **Katharine Robb

---

## [Decision Letter · Decision Letter 1]

17 Mar 2025

PGPH-D-24-02175R1

From the Ground Up: Stakeholder Perspectives on Housing Interventions to Reduce Environmentally Mediated Infections

Dear Dr. Benjamin-Chung,

Thank you for submitting your manuscript to PLOS Global Public Health. After careful consideration, we feel that it has merit but does not fully meet PLOS Global Public Health’s publication criteria as it currently stands. Therefore, we invite you to submit a revised version of the manuscript that addresses the points raised during the review process.

We request you to kindly address the minor concerns raised by Reviewer 3. 

We look forward to receiving your revised manuscript.

Kind regards,

Annesha Sil, Ph.D.

Staff Editor

PLOS 

Journal Requirements:

Additional Editor Comments (if provided):

Reviewers' comments:

Reviewer's Responses to Questions

**Comments to the Author**

1. If the authors have adequately addressed your comments raised in a previous round of review and you feel that this manuscript is now acceptable for publication, you may indicate that here to bypass the “Comments to the Author” section, enter your conflict of interest statement in the “Confidential to Editor” section, and submit your "Accept" recommendation.

Reviewer #1: All comments have been addressed

Reviewer #3: All comments have been addressed

2. Does this manuscript meet PLOS Global Public Health’s publication criteria? Is the manuscript technically sound, and do the data support the conclusions? The manuscript must describe methodologically and ethically rigorous research with conclusions that are appropriately drawn based on the data presented.

Reviewer #1: Yes

Reviewer #3: Yes

3. Has the statistical analysis been performed appropriately and rigorously?

Reviewer #1: N/A

Reviewer #3: N/A

4. Have the authors made all data underlying the findings in their manuscript fully available (please refer to the Data Availability Statement at the start of the manuscript PDF file)?

Reviewer #1: Yes

Reviewer #3: Yes

5. Is the manuscript presented in an intelligible fashion and written in standard English?

Reviewer #1: Yes

Reviewer #3: Yes

6. Review Comments to the Author

Reviewer #1: Thank you for responding to all of the reviewers' comments so thoroughly and so thoughtfully. I am satisfied that the paper is greatly improved and ready for publication. It will be a useful contribution to housing and health literature.

Reviewer #3: The authors have adequately addressed the comments.

However, I would suggest that they include in the manuscript more of the justification around lack of government representatives as key informants. In their reviewer response (but not in the manuscript) they hypothesize that it was difficult to identify relevant government actors because "housing and health interventions –besides ones made for sanitation, household energy or bednets – are not a focus of government programs." This is an important perspective - perhaps one which they could also support with evidence - as it also has implications for how housing interventions to improve health.

Finally, a very minor comment but the section on Swachh Bharat uses the word estimate/estimated many times - I suggest finding some alternative phrasing.

7. PLOS authors have the option to publish the peer review history of their article (what does this mean?). If published, this will include your full peer review and any attached files.

**Do you want your identity to be public for this peer review?** For information about this choice, including consent withdrawal, please see our Privacy Policy.

Reviewer #1: **Yes: **Nina Lansbury

Reviewer #3: **Yes: **Dr. Katharine Robb

---

## [Editor Report · Decision Letter 2]

26 Mar 2025

From the Ground Up: Stakeholder Perspectives on Housing Interventions to Reduce Environmentally Mediated Infections

PGPH-D-24-02175R2

Dear Dr. Benjamin-Chung,

We are pleased to inform you that your manuscript 'From the Ground Up: Stakeholder Perspectives on Housing Interventions to Reduce Environmentally Mediated Infections' has been provisionally accepted for publication in PLOS Global Public Health.

Best regards,

Julia Robinson

Executive Editor